# Core Genome Sequencing Analysis of *E. coli* O157:H7 Unravelling Genetic Relatedness among Strains from Cattle, Beef, and Humans in Bishoftu, Ethiopia

**Fanta D. Gutema** [1,2], **Lieven De Zutter** [3], **Denis Piérard** [4], **Bruno Hinckel** [5], **Hideo Imamura** [5], **Geertrui Rasschaert** [6], **Reta D. Abdi** [7], **Getahun E. Agga** [8] **and Florence Crombé** [4,*]

1 Department of Occupational and Environmental Health, University of Iowa, Iowa City, IA 52242, USA
2 Department of Microbiology, Immunology and Veterinary Public Health, Addis Ababa University, Bishoftu P.O. Box 34, Ethiopia
3 Department Translational Physiology, Infectiology and Public Health, Faculty of Veterinary Medicine, Ghent University, 9820 Merelbeke, Belgium
4 Laboratory of Microbiology and Infection Control, Department Clinical Biology, Belgian National Reference Centre for STEC/VTEC, Vrije Universiteit Brussel (VUB), Universitair Ziekenhuis Brussel (UZ Brussel), 1090 Brussels, Belgium
5 Brussels Interuniversity Genomics High throughput Core (BRIGHTcore) Platform, Vrije Universiteit Brussel (VUB), Universitair Ziekenhuis Brussel (UZ Brussel), Laarbeeklaan, 1090 Brussels, Belgium
6 Technology and Food Science Unit, Flanders Research Institute for Agriculture, Fisheries and Food, 9090 Melle, Belgium
7 Department of Veterinary Biomedical Sciences, College of Veterinary Medicine, Long Island University, Greenvale, NY 11548, USA
8 Food Animal Environmental Systems Research Unit, U. S. Department of Agriculture, Agricultural Research Service, 2413 Nashville Road, B-5, Bowling Green, KY 42101, USA
* Correspondence: florence.crombe@uzbrussel.be

**Abstract:** *E. coli* O157:H7 is a known Shiga toxin-producing *Escherichia coli* (STEC), causing foodborne disease globally. Cattle are the main reservoir and consumption of beef and beef products contaminated with *E. coli* O157:H7 is an important source of STEC infections in humans. To emphasize the cattle-to-human transmission through the consumption of contaminated beef in Bishoftu, Ethiopia, whole-genome sequencing (WGS) was performed on *E. coli* O157 strains isolated from three sources (cattle, beef, and humans). Forty-four *E. coli* O157:H7 isolates originating from 23 cattle rectal contents, three cattle hides, five beef carcasses, seven beef cuts at retail shops, and six human stools in Bishoftu between June 2017 and May 2019 were included. This study identified six clusters of closely related *E. coli* O157:H7 isolates based on core genome multilocus sequence typing (cgMLST) by targeting 2513 loci. A genetic linkage was observed among the isolate genomes from the cattle rectal contents, cattle hides, beef carcasses at slaughterhouses, beef at retail shops, and human stool within a time frame of 20 months. All the strains carried practically the same repertoire of virulence genes except for the *stx*2 gene, which was present in all but eight of the closely related isolates. All the strains carried the *mdf*A gene, encoding for the MdfA multi-drug efflux pump. CgMLST analysis revealed genetically linked *E. coli* O157:H7 isolates circulating in the area, with a potential transmission from cattle to humans through the consumption of contaminated beef and beef products.

**Keywords:** *E. coli* O157:H7; whole-genome sequencing; cgMLST; transmission; virulence genes; antimicrobial resistance

## 1. Introduction

Shiga toxin-producing *Escherichia coli* (STEC) are zoonotic pathotypes of diarrheagenic *E. coli*. They are characterized by the production of Shiga toxins causing enteric infections ranging from mild self-limited diarrhea to severe infections such as bloody diarrhea, hemorrhagic colitis, and even life-threatening conditions such as hemolytic uremic syndrome

(HUS) and thrombotic thrombocytopenic purpura (TTP) in humans [1]. There are more than 470 serotypes of STEC [2]. Among these serotypes, *E. coli* O157:H7 is the most widely known and well-studied serotype reported as a cause of foodborne illness [3]. Currently, there are also increasing reports on the significance of non-O157:H7 *E. coli* serotypes linked with foodborne outbreaks and severe infections in humans [4–6].

According to the 2010 Foodborne Disease Burden Epidemiology Reference Group (FERG) database, the burden per 100,000 STEC in Ethiopia was more than 10 times lower than the burden at the global level (0.2 [0.09–0.05] foodborne disability-adjusted life years per 100,000 population), which was more than two orders of magnitude lower than the burden of *Campylobacter* spp., enterotoxinogenic *E. coli*, and non-typhoidal *Salmonella enterica* [7]. As the documented burden of STEC in Africa was very low at the time of the study, the burden estimates could not be updated for 2017 [7].

Although *E. coli* O157 resides in the gut of different animals, cattle are recognized as the main reservoir of the pathogen [1,8]. In cattle, colonization by *E. coli* O157 is considered asymptomatic, and hence, cattle can serve as a source of human infections [9]. Human exposure by direct contact with infected cattle or with their feces is also an important route of transmission [1,10,11]. Further, contaminated meat, dairy products, vegetables, and water contaminated by animal feces are also common sources of human infection [12–14]. In particular, the consumption of contaminated raw or undercooked beef and beef products is an important risk factor, and several STEC infections were attributed to these foods globally [15]. For instance, in the EU/EEA, based on outbreaks reported to the European Food Safety Authority (EFSA) from 2012 to 2017, the consumption of bovine meat and products was identified as a major source of STEC, attributing to 24% of STEC outbreaks [16]. In the United States, among 466 reported STEC outbreaks between 2010 and 2017 affecting 4769 persons, 20% of the outbreaks were linked to beef, and 71% of the outbreaks were caused by *E. coli* O157 [14]. Recently, in Ethiopia, a Structured Expert Elicitation study attributed about 60% of the burden of STEC in beef to red meat and about 31% to beef consumed raw [17].

In developing countries like Ethiopia, information on the transmission pathways and the zoonotic importance of *E. coli* O157 infection is lacking. In our previous studies, we observed the probable relatedness among *E. coli* O157 strains isolated from (1) cattle rectal contents, beef, and humans and (2) cattle rectal contents and/or contaminated hides and carcasses based on the pulsed-field gel electrophoresis (PFGE) typing method [18,19]. The studies signaled the occurrence and spread of the pathogen in the cattle–beef–human continuum with a potential to cause diarrheal illness in exposed individuals. PFGE is a commonly used genotyping method for *E. coli* O157, especially in outbreak investigations [20]. However, it cannot provide a true phylogenetic measure and does not differentiate strains to the same degree as DNA fragments are separated according to their size, regardless of the gene sequences [21]. It also does not provide genetic information on the virulence potential and resistance genes carried by pathogens that can be achieved by whole-genome sequencing (WGS) [20]. WGS is recognized as the best molecular subtyping method due to its high discrimination power over other methods [22]. Currently, it is becoming a gold standard method for the global surveillance of foodborne diseases in PulseNet international network countries [23]. The objective of this study was to sequence the genome of *E. coli* O157 strains obtained from cattle, beef, and humans to determine the phylogenetic relationship among the strains. This would indicate the potential transmission of *E. coli* O157 from cattle to humans via the consumption of contaminated beef in the study area, where the consumption of raw and undercooked beef is common. The information can be used to inform national policymakers about the surveillance of foodborne pathogens along beef production and supply chains, thereby designing intervention measures to ensure beef food safety.

## 2. Materials and Methods

### 2.1. Sources and Detection of E. coli O157

We characterized 44 *E. coli* O157 isolates previously identified during a study period from June 2017 to May 2019. The isolates originated from 23 cattle rectal contents, 3 cattle hides, 5 beef carcasses, 7 beef cuts at retail shops, and 6 human stools. Cattle rectal contents were collected before slaughter (n = 240) and after slaughter (n = 70), and hide swabs were collected from 70 carcasses with hides-on, i.e., before de-hiding at slaughterhouses. Isolates from beef and beef carcasses originated from 127 beef cuts collected from all available retail shops in Bishoftu Town and 70 beef carcasses at slaughterhouses, respectively. The details of sampling procedures and laboratory methods for the detection and characterization of *E. coli* O157 isolates are described in our previous studies [18,19].

### 2.2. Pulsed-Field Gel Electrophoresis

The fingerprints obtained from our previous studies [18,19] were reanalyzed in order to enable pulsotype matching. The fingerprints were grouped according to their similarity with BioNumerics v.8.1 (Applied Maths, BioMérieux, Sint-Martens-Latem, Belgium) using the Pearson coefficient and unweighted pair group method using arithmetic averages with an optimization of 2%. Pulsotypes were assigned based on the difference of at least one band in the fingerprints and indicated by Roman numerals.

### 2.3. Whole-Genome Sequencing

Genomic DNA was extracted from pure cultures of *E. coli* O157 isolates grown overnight on a Sorbitol MacConkey Agar (Neogen, Lansing, MI, USA) by using a Maxwell RSC Cell DNA purification kit (Promega Corporation, Madison, WI, USA) according to the manufacturer's instructions. The purity and quantity of the genomic DNA were measured, respectively, with a NanoDrop 2000 C (Thermo Fisher Scientific, Waltham, MA, USA) and a Qubit 2.0 Fluorometer (Thermo Fisher Scientific, Waltham, MA, USA) with a Qubit dsDNA BR assay kit.

Fragmentation of 500 ng of genomic DNA was carried out using the NEBNext® Ultra™ II FS module. Sequencing libraries, with an insert size of on average 550 bp, were prepared using a KAPA Hyper Plus kit (Kapa Biosystems, Wilmington, USA) and a Pippin Prep (Sage Science, Beverly, MA, USA) size selection with a CDF1510 1.5% agarose dye-free cassette. To avoid PCR bias, the PCR amplification step was omitted, and every sample was assigned an in-house truseq style adapter with a unique dual-indexed 8-bp barcode. After equimolar pooling, libraries were sequenced on a Novaseq 6000 instrument (Illumina, San Diego, CA, USA) using a NovaSeq 6000 SP Reagent kit (500 cycles) generating $2 \times 250$ bp reads. For this, the library was denatured and diluted according to the manufacturer's instructions. A 1% PhiX control library was included in each sequencing run.

The raw reads were uploaded and de novo assembled, using SPAdes v.3.7.1, in BioNumerics v.8.1. Sequence quality was assessed using the quality metrics incorporated in BioNumerics v.8.1. The major quality parameters are summarized in the Supplementary Materials, Table S1.

### 2.4. Core Genome MLST Analysis

The assembled sequencing data was analyzed using the *Escherichia/Shigella* cgMLST typing scheme in BioNumerics v.8.1 (core Enterobase). This scheme consists of 2,513 loci. Both assembly algorithms were used for allele calling, i.e., the assembly-free k-mer-based approach using the raw reads and the assembly-based BLAST approach. The default settings were used for both the assembly-free and assembly-based algorithms. The quality of the assembly-free and the assembly-based allele calls were verified using the quality statistics window in BioNumerics. The MLST profile of each isolate was determined using the PubMLST (Achtman) allele mapping experiment incorporated in BioNumerics. Minimum spanning trees (MSTs) were generated from the cgMLST allelic profiles of the isolates using the predefined template "MST for categorical data" in BioNumerics. Branch

lengths reflect the number of allele differences (ADs) between the allelic profiles of the isolates in the connected nodes. For clustering, the partitioning algorithm was used. The partitioning threshold was set to 4, which results in MST clusters with less than 5 ADs, highlighted in grey.

### 2.5. Comparison of E. coli O157:H7 in EnteroBase

The raw reads of all 44 *E. coli* genomes were also uploaded and automatically assembled in the public genome database, EnteroBase. All genome assemblies were subsequently compared to the available *E. coli* genomes using hierarchical clustering of cgMLST (HierCC) at different levels of resolution, ranging from HC0 (hierarchical clusters consisting of identical genomes with no AD) to HC200 (hierarchical clusters consisting of genomes with up to 200 ADs) [24].

### 2.6. In Silico Identification of Genes Linked to Serotype, Virulence, Antibiotic Resistance, and Plasmids

The *E. coli* genotyping tool, available in BioNumerics v.8.1, was used to predict *E.coli* serotypes, virulence gene profiles, acquired resistance genes, point mutations, and plasmids starting from the genome assemblies. The presence of virulence and resistance genes was determined with a minimum % identity (ID) threshold of 85% and a minimum length for coverage of 60%.

## 3. Results

### 3.1. Pulsed-Field Gel Electrophoresis

The 44 *E. coli* O157 isolates were grouped into 15 pulsotypes (I-VX) (Figure 1 and Supplementary Materials, Table S2), matching two pulsotypes from each study to pulsotypes VI (pulsotype D [18] and B [19]) and VIII (pulsotype C in both studies). As observed in our previous study, three pulsotypes (VI, VII, and XI) contained isolates from the three sources (cattle, beef, and humans), pulsotype IX contained isolates from two sources (beef and human), and the remaining pulsotypes contained one or more isolates from only one source [18,19]. Pulsotypes VI and VII contained *stx*2-negative and *stx*2-positive isolates, respectively, that were closely related but not identical [18,19]. Among the isolates obtained from the same animal (n = 3), genetic relatedness was observed only between isolates obtained from a hide and a carcass (the hind leg) swab of one animal sampled at the municipal slaughterhouse [19].

### 3.2. Core Genome MLST Analysis

Six clusters were identified based on the cgMLST analysis of the 44 *E. coli* O157 isolate genomes, with sequence type (ST) 11: Cluster 1 (n = 11 isolate genomes; range 0–2 ADs), Cluster 2 (n = 10; range 0–3), Cluster 3 (n = 9; range 0–1), Cluster 4 (n = 3; no allelic difference), Cluster 5 (n = 5; range 0–2), and Cluster 6 (n = 4; range 0–1) (Figure 2, Supplementary Materials, Table S2). Accordingly, only two isolate genomes (MB5R and PE1R) retrieved from the cattle rectal contents did not belong to any of the identified clusters. It is to be noted, however, that one of these two isolate genomes (PE1R) is possibly closely related to the isolate genomes from Cluster 3 (range: 7–8). The number of allelic differences between the clusters was at least 84 alleles.

Interestingly, based on the cgMLST analysis, three molecular links were observed between the *E. coli* O157 strains (Figure 2, Supplementary Materials, Table S2). First, four clusters included genomes of the strains isolated from the cattle rectal content/hide and beef carcass (Clusters 1, 3, 5, and 6). Second, two clusters included genomes of the strains isolated from the beef carcass and beef (Clusters 1 and 5). Third, four clusters included genomes of the strains isolated from the beef and human stool (Clusters 1, 2, 4, and 5). Consequently, four clusters included genomes isolated from all three sources (cattle, beef, and humans) (Clusters 1, 2, 4, and 5).

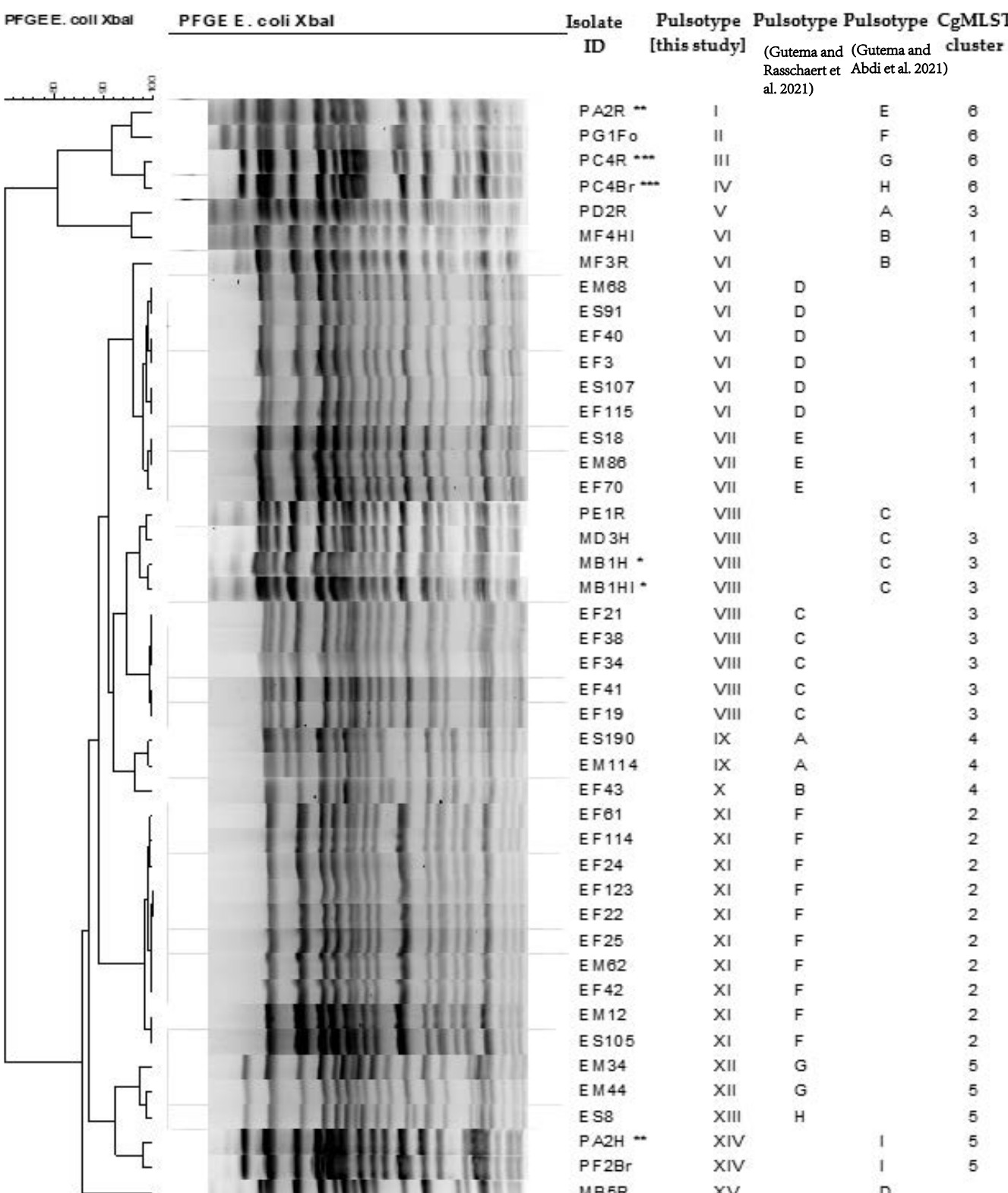

**Figure 1.** PFGE patterns of all *E. coli* O157 isolates included in our study, with indication of the pulsotypes and the cgMLST clusters. *, **, and ***: isolates from the same animal [18,19].

Remarkably, as depicted in Figure 3, the isolate genomes were highly related over the sampling period (Supplementary Materials, Table S2). Cluster 1, for example, includes strains retrieved from the cattle rectal contents at the municipal slaughterhouse between

June and July 2017 and in February 2019, approximately 20 months apart. In addition, the genomes of the human stool isolates (ES18, ES91, and ES107), retrieved up to 5 months apart in 2018, were closely related to the first cattle isolate (EF3), identified in June 2017 within Cluster 1. Similarly, Cluster 3 includes strains retrieved from the cattle rectal content (EF19, EF21, EF34, EF38, and EF41) at the municipal slaughterhouse in July 2017 but also one isolate retrieved from the cattle rectal content (PD2R) at the private slaughterhouse in January 2019. Yet, as mentioned earlier, another isolate retrieved from the cattle rectal content (PE1R) at the private slaughterhouse in January 2019 could rather be considered an evolved clone, differing up to eight alleles from the isolate genomes from Cluster 3 (Figures 2 and 3).

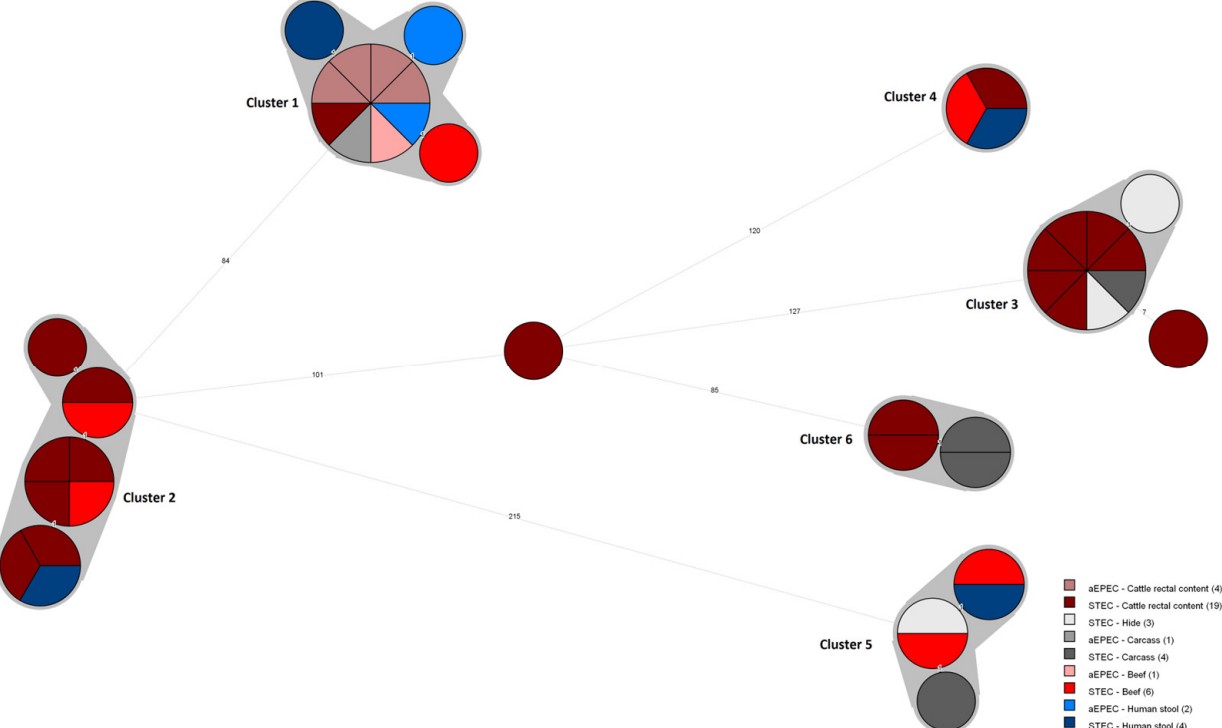

**Figure 2.** Minimum spanning tree based on cgMLST allelic profiles of 44 *E. coli* genomes built from cgMLST analysis (core Enterobase, BioNumerics v.8.1). Nodes are color-coded per isolate pathotype, originating specimen, and their respective numbers, as labeled. Numbers of allelic differences are indicated on the lines connecting the various core cgMLSTs.

Two isolates recovered from the hide and carcass of the same animal (MB1H and MB1HI) and the rectal content and carcass samples of the same animal (PC4R and PC4Br) were highly similar and belonged to the same cluster (Cluster 3 and Cluster 6, respectively), with only a 0–1 AD. Yet, two isolates (PA2R and PA2H) belonging to different clusters were also retrieved from the rectal content and hide samples of the same animal (Supplementary Materials, Table S2).

### 3.3. Comparison of E. coli O157:H7 Genomes in EnteroBase

All isolate genomes included in this study were assigned to cluster HC200_63 (Supplementary Materials, Table S3), with up to 200 ADs, which also includes isolates from humans and other animals circulating worldwide.

All isolate genomes from Cluster 1 belonged to cluster HC100_37630, which also included one genome from a human clinical case in the United Kingdom (Table 1). No other isolate genomes were assigned to cluster HC50_205870, which means that no other isolate genomes uploaded in EnteroBase have links no more than 50 alleles apart.

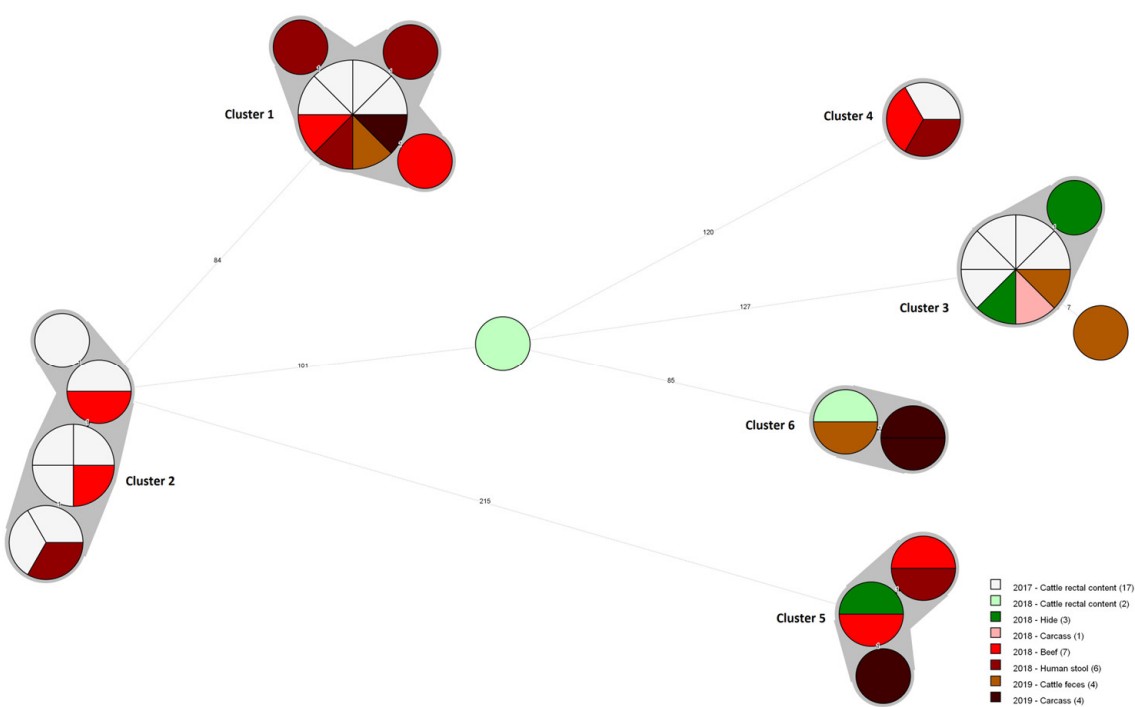

**Figure 3.** Minimum spanning tree based on cgMLST allelic profiles of 44 *E. coli* genomes built from cgMLST analysis (core Enterobase, BioNumerics v.8.1). Nodes are color-coded per year of isolation, originating specimen, and their respective numbers, as labeled. Numbers of allelic differences are indicated on the lines connecting the various cgMLSTs.

**Table 1.** *E. coli* isolate genomes closest to the genomes included in this study.

| HC100 [1] | HC50 | HC20 | HC10 | HC5 | Origin | Sample Type | Country of Isolation | Collection Year |
|---|---|---|---|---|---|---|---|---|
| 1022 | 15671 | 147247 | 147247 | 147247 | Cattle | Beef tartare, carcass | Belgium | 2005 |
| | 15671 | 147247 | 147247 | 147247 | Cattle, human | Hide, carcass, beef, stool | Ethiopia | 2018, 2019 |
| 16335 | 205844 | 205844 | 205844 | 205844 | Cattle | Rectal content, hide, carcass | Ethiopia | 2017, 2018, 2019 |
| | 205844 | 205844 | 205844 | 205846 | Cattle | Rectal content | Ethiopia | 2019 |
| | 205844 | 205844 | 205844 | 211075 | Human | NS | United Kingdom | 2022 |
| 30437 | 30437 | 30437 | 30437 | 30437 | Human | NS | United Kingdom | 2016 |
| | 205843 | 205843 | 205843 | 205843 | Cattle, human | Rectal content, beef, stool | Ethiopia | 2017, 2018 |
| | 205845 | 205845 | 205845 | 205845 | Cattle | Rectal content, carcass | Ethiopia | 2018, 2019 |
| | 205847 | 205847 | 205847 | 205847 | Cattle | Rectal content | Ethiopia | 2018 |
| | 205871 | 205871 | 205871 | 205871 | Cattle, human | Rectal content, beef, stool | Ethiopia | 2017, 2018 |
| 37630 | 37630 | 37630 | 37630 | 37630 | Human | NS | United Kingdom | 2016 |
| | 205870 | 205870 | 205870 | 205870 | Cattle, human | Rectal content, carcass, beef, stool | Ethiopia | 2017, 2018, 2019 |

[1] Hierarchical clusters (HCs) with genomes up to 100 allele differences.

All isolate genomes from Clusters 2, 4, and 6 and the non-cluster isolate, MB5R, belonged to cluster HC100_30437, which also included isolate genomes from human cases retrieved from the United Kingdom (Table 1). No other isolate genomes were assigned to the HC50 clusters, HC50_205871, HC50_205843, HC50_205845, and HC50_20847, respectively. A human isolate genome from the United Kingdom was assigned to cluster HC10_205844, in 2022, after the upload of the non-cluster isolate, MB5R.

All isolate genomes from Cluster 5 were assigned to cluster HC100_1022, including genomes of isolates from humans and animals circulating worldwide; HC50_15671, including isolate genomes from humans and animals circulating in Europe; and HC20_147247, including two Belgian isolate genomes from cattle carcasses and beef tartare (*filet Americain*) (Table 1). Remarkably, the isolate genomes from Cluster 5 and the Belgian isolate genomes were assigned to HC5_147247, which means that all the strains in this cluster have links no more than five alleles apart. More surprisingly, two isolate genomes from Cluster 5 and the Belgian isolate genome from a cattle carcass were even indistinguishable (HC0_147247) (the data are not shown).

### 3.4. In Silico Identification of Genes Linked to Serotype, Virulence, Antibiotic Resistance, and Plasmids

All isolates were confirmed as *E. coli* 0157:H7 by in silico serotyping.

It was observed that all the strains carried practically the same repertoire of virulence genes encoding for adherence factors (*eae*, *iha*, and *tir*), type III translocated proteins (*esp*A, *esp*B, *esp*F, *esp*J, *etp*D, *nle*A, *nle*B, *nle*C, and *tcc*P), outer membrane proteins (*chu*A, *omp*T, and *tra*T), toxins (*ast*A, *ehx*A, and *tox*B), and other pathogenicity-related factors (*gad*, *iss*, *kat*P, and *ter*C) (Supplementary Materials, Table S4). None of the strains carried *stx*1. All but eight strains carried *stx*2, i.e., subtypes *stx*2a (5/44) and *stx*2c (31/44).

All of the strains carried the attaching and effacing intimin gene (*eae*) but lacked the plasmid-mediated bundle-forming pilus gene (*bfp*A), classifying the eight *stx*-negative *E. coli* strains as atypical enteropathogenic *E. coli* (aEPEC) (*eae*+ and *bfp*-).

Remarkably, Cluster 1 encompasses *stx*2-positive and *stx*2-negative strains in contrast to the other clusters. The genes coding for the *stx*2c subunits A and B were detected in three out of the eleven *E. coli* Cluster 1 isolate genomes. The *stx*2-positive strains were isolated from the beef (EM86), cattle rectal content (EF70), and human stool (ES18) on different sampling moments (Figure 2 and Supplementary Materials, Table S2). Similarly, the *stx*-negative strains were isolated from all three sources on different sampling moments (Figure 3 and Supplementary Materials, Table S2).

None of the *E. coli* O157:H7 isolate genomes harbored acquired antimicrobial resistance genes, except the *mdf* A gene that encodes for the MdfA multi-drug efflux pump, occurring in all of the isolates. Resistance-conferring point mutations were not detected.

The plasmids IncFIB(AP001918) and IncFII were identified among all *E. coli* O157:H7 isolate genomes.

### 4. Discussion

To the best of our knowledge, this is the first work that describes the genetic linkage between *E. coli* O157 in cattle, beef carcasses, beef cuts, and human stool samples originating from diarrheic patients using WGS technology in Ethiopia.

CgMLST analysis of 44 previously identified *E. coli* O157 isolates revealed six clusters of closely related isolate genomes (<5 AD) within a time frame of 20 months. Interestingly, four out of the six identified clusters included genomes of strains isolated from all three sources (cattle, beef, and humans), inferring the possible transmission of *E. coli* O157 from cattle to humans through contaminated beef in the study area. The consumption of beef and beef products contaminated at slaughterhouses and meat processing plants has been associated with *E. coli* O157 outbreaks in numerous reports [11,15,25,26]. Based on our results, however, it was not possible to confirm that human infection was caused by the consumption of contaminated raw beef, as a direct epidemiological link was missing. Yet, the consumption of raw beef in the form of steak (*kurt*) dipped in plant-based spices or beef tartare (*kitfo*) made from raw minced beef is very common in Ethiopia [18]. Despite the limited number of isolates recovered from diarrheic patients, all of the clinical isolate genomes belonged to one of the four source-overlapping clusters. Moreover, clonal strains were retrieved from the cattle rectal content, beef carcasses at slaughterhouses, and beef at retail shops, showing the occurrence, transmission, and survival of *E. coli* O157:H7 strains

along the entire supply chain (the slaughterhouse to retail shops), raising concern of beef safety in the study area.

The presented results expand upon our previous results obtained with PFGE [18,19]. Still, a small fraction of the isolates, which were considered similar but not identical by PFGE, were clonal by WGS. The present analysis underlines, once more, the importance of WGS as a high-resolution molecular tool in tracking the sources of food contamination and foodborne outbreak investigations [23].

In this study, the inclusion of *E. coli* isolates collected within a time frame of 20 months improved our ability to find clonal isolates that overlap among all three sources (cattle, beef, and humans). Moreover, over time, clonal *E. coli* O157:H7 strains circulating in the area were identified. The cluster representative isolates differed, with at least 84 alleles from each other, and are, therefore, unrelated [27]. This indicates the persistence and widespread dissemination of resident *E. coli* O157:H7 strains (*stx*2- or *stx*2+, and *eae*+) in the study area, with a possible transmission between animals and humans, raising concern for public health intervention. *E. coli* O157 has the ability to survive in many adverse ecological conditions and persists in the environment, such as in the soil, manure, water, and the animal's immediate environment for several days, making it more complex to determine the transmission pathways apart from the possible direct transfer from cattle to humans via the consumption of beef and beef products [28,29]. Future studies should consider environmental-based transmissions and other food systems to identify the mediating factors and determine the predominant transmission pathways to design and implement tailored preventions and control measures.

It is to be noted that the samples were collected from one city, which may not represent the strains circulating in the whole country. Only the situation in this area was described. As the national prevalence estimation of beef contamination is estimated at 6%, it is interesting to unravel this [30].

Looking at the related genomes available in EnteroBase, the isolate genomes from this study were dispersed among other HC200_63 isolate genomes from the rest of the world; however, the majority of the isolate genomes from this study belonged to a new genomic HC50 sub-cluster. Indeed, all of the study isolate genomes fell into seven HC50 sub-clusters (50 ADs), comprising the isolate genomes from this study only, except for the isolate genomes from Cluster 5 (HC50_15671). The latter was even assigned to cluster HC5_147247, including the isolate genomes from this study and two Belgian isolate genomes from cattle. We were even more surprised to find that two out of the five isolate genomes from Cluster 5 were identical to one of the Belgian isolate genomes (HC0_147147), although the samples were collected 13 years apart. At present, we have no explanation for these results. Yet, WGS analysis should always be interpreted in an epidemiological context, certainly with epidemics occurring overseas and being able to be imported through international trade or travel [31]. It is to be noted that only approximately 3.5% ($n_{\text{Africa}}$ = 8506; $n_{\text{Total}}$ = 239,796) of the publicly available *E. coli/Shigella* assembled genomes in EnteroBase comes from Africa (consulted on 12 December 2022). Therefore, due to the limited available genomic data, it is impossible to ensure that no other related isolates are circulating in the regions or countries neighboring the study area.

In the present study, we found an almost uniform repertoire of virulence genes amongst all of the *E. coli* O157:H7 isolates, with some exceptions. The main difference was in the presence of the *stx*2 gene, which was present in all but eight closely related isolates. These results are of importance, as no virulence data are published based on WGS in Ethiopia. Major virulence factor-encoding genes (i.e., *stx*2+, *eae*+, and *ehxA*+) were identified among the majority of our isolates, indicating their pathogenic potential for humans. Indeed, the presence of the *stx*2 gene has been associated with severe disease [32]. Subtypes *stx*2a and *stx*2d are associated with a high risk of HUS development, while the other *stx*2 subtypes, including the *stx*2c and *stx*1 subtypes, are associated with medium and low HUS risk, respectively [27,28]. The *eae* gene, encoding for intimin, an outer membrane protein that facilitates intimate attachment to the intestinal epithelial cells, also plays a role

in STEC pathogenicity. The presence of both the *stx*2 and *eae* genes is associated with a greater probability of triggering severe disease [28,32]. Moreover, other virulence-related genes involved in different mechanisms (i.e., *ehx*A, *esp*B, *esp*F, *esp*J, and *tir*) were described to be associated with highly virulent STEC strains and the potential of causing disease [26,33].

All *stx*-negative strains were classified as aEPEC (*eae*+, *bfp*-). Interestingly, all aEPEC isolates were closely related to the STEC isolates within one source-overlapping cluster, suggesting that the *stx*-negative strains might have lost or acquired the *stx* gene during infection [34,35]. It has been shown that aEPEC and STEC co-exist in vivo, representing a highly dynamic system that can convert in both directions by the loss and gain of Stx-encoding phages [36–38]. Moreover, the loss and gain of Stx-encoding phages have been associated with similar but not identical PFGE patterns [36], which was also the case in our previous study [18]. Conversely, the *stx*-negative strains might have lost the *stx* gene during isolation or the subculture [36,37]. Consequently, the laboratory diagnosis of *stx*-negative strains should be considered, as these strains have been associated with diarrheal disease and outbreaks [39,40].

In support of our previous findings, all *E. coli* O157:H7 isolates included in the present study did not carry any antibiotic resistance genes or point mutations, encoding for the 14 drugs we previously tested [18,19]. All of them carried the *mdf*A gene encoding for the non-specific MdfA multi-drug efflux pump [41]. This gene seems to be commonly present among *E. coli* isolates independent of the source of isolation [42,43].

## 5. Conclusions

The cgMLST analysis showed a genetic linkage among *E. coli* O157:H7 strains from cattle, beef carcasses, beef, and humans, suggesting a possible transmission from cattle to humans through contaminated beef. Moreover, the isolate genomes were closely related over the sampling period, which indicates the persistence and widespread dissemination of several resident strains in the area. Efforts toward the use of WGS technology are highly required for the surveillance and tracking of the transmission pathways of foodborne pathogens and antimicrobial resistance between animals and humans in Ethiopia.

**Supplementary Materials:** The following supporting information can be downloaded at: https://www.mdpi.com/article/10.3390/microbiolres14010013/s1, Table S1: Quality statistics of all *E. coli* isolate genomes sequenced in this study; Table S2: Relevant microbiological and host-associated information of all *E. coli* isolates included in our study; Table S3: Hierarchical clustering of core genome multilocus sequence typing (cgMLST) (HierCC) results obtained from EnteroBase for all *E. coli* isolate genomes, with indication of the cgMLST clusters obtained from BioNumerics; Table S4: Whole-genome sequencing (WGS)-based virulence-associated gene profiles for all *E. coli* isolates included in our study, with indication of the core genome multilocus sequence typing (cgMLST) clusters and pathotype. References [18,19] are cited in the supplementary materials.

**Author Contributions:** Conceptualization, F.D.G., R.D.A., G.E.A. and L.D.Z.; formal analysis, F.C., G.R., B.H. and H.I.; investigation, F.D.G., L.D.Z., G.R., R.D.A. and G.E.A.; resources, F.D.G., G.R. and D.P.; data curation, F.D.G.; writing—original draft preparation, F.D.G. and F.C.; writing—review and editing, L.D.Z., D.P., G.R., R.D.A. and G.E.A..; visualization, F.C. and G.R.; supervision, L.D.Z.; project administration, F.D.G.; funding acquisition, D.P. All authors have read and agreed to the published version of the manuscript.

**Funding:** Previous research was supported by Addis Ababa University and Ghent University under the Special Research Fund (BOF) program for developing countries (Scholarship code 01 W03916) [18,19]. The WGS research received no external funding.

**Institutional Review Board Statement:** The study was conducted in accordance with the Declaration of Helsinki and approved by the Institutional Review Board of Addis Ababa University (Ref: VM/ERC/06/05/09/2017), the Ministry of Science and Technology of Ethiopia (Ref:3/10/006/2018), and University Hospital of Gent, Belgium (Ref:2017/0612).

**Informed Consent Statement:** Informed consent was obtained from all subjects involved in the study.

**Data Availability Statement:** All *E. coli* O157:H7 genomes are available online in the EnteroBase *Escherichia/Shigella* database (https://enterobase.warwick.ac.uk). The isolates can be obtained from EnteroBase using the 'Search Strains' parameter and under 'Strain Metadata', selecting the 'Barcode' option, and entering the EnteroBase barcodes in the 'Value' box. The EnteroBase barcodes are provided in the Supplementary Materials, Table S3. All supporting data and protocols have been provided within the article or through supplementary data files. Supplementary Materials are available with the online version of this article.

**Acknowledgments:** We would like to thank the scientists and technicians of the Brussels Interuniversity Genomics High Throughput core (BRIGHTcore; funded by the VUB grant OZR2434, ULB and « the Foundation against Cancer » grant 2016-021, UZ Brussel and Hôpital Erasme; www.brightcore.be). Mention of trade names or commercial products in this publication is solely to provide specific information and does not imply recommendation or endorsement by the U.S. Department of Agriculture. USDA is an equal opportunity provider and employer.

**Conflicts of Interest:** The authors declare no conflict of interest.

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
