# Peer review of "Core Genome Sequencing Analysis of E. coli O157:H7 Unravelling Genetic Relatedness among Strains from Cattle, Beef, and Humans in Bishoftu, Ethiopia"

_2036-7481, doi:10.3390/microbiolres14010013_

Round 1

Reviewer 1 Report

Authors provided a decent justification by introducing E. coli O157:H7, its widespread  especially in the US and Europe followed by previous reports from their  studies and performed WGS on E. coli. Authors considered different sources to isolate e.coli O157 strains from three different sources. Further authors performed cluster analysis via cgMLST.  

If there are any previous reports on the burden of foodborne pathogens in Ethiopia, please provide this information. Also, in the abstract please provide the total representative sample size where these 44 E. coli isolates were collected.

Authors mentioned under purpose of the study-to ascertain the hypothesis of ecoli O157 transmission from cattle to humans via the consumption of contaminated beef in countries…It is a great hypothesis to assume. However, 44 e.coli genomes collected from only one city is significantly less to draw such conclusions. Therefore, I would suggest rephrasing the sentences in the abstract and discussion. Discussion is also well written. Overall, it is a great study which is current and required for surveillance studies using WGS. 

Reviewer 2 Report

the authors identified 44 E. coli O157:H7 isolates from cattle, beef, and humans in in Ethiopia. WGS was performed and the strains were analysed by PFGE, cgMLST, Virulence and antimicrobial resistance. The manuscript was well-written and it is interesting. I just have a minor comment related to the chromosomal point mutations (GyrA, and ParC) against ciprofloxacin. Also, It is very interesting to identify the plasmid content of these strains by PlasmidFinder available at the Center for Genomic Epidemiology (CGE) and to identify whether the virulence genes were chromosomal or plasmid borne. 

Reviewer 3 Report

I found this manuscript very interesting in the field. The complementation of the PFGE and whole genome sequencing is quite relevant. However, the genomic analysis seems quite superficial and must be improved in my opinion.

My main concern is that it relies only on the analysis of cgMLST. I think that a previous analysis of ANI and/or smash distances must be analyzed. High ANI  values would identify isolates that belong to the same circulating strain, which could reveal the actual diversity within the 44 analyzed isolated.

The work would also benefit from a more detailed description of the core genomes. 

Other questions that may be addressed are the presence of other mobile elements in these strains that have been related to pathogenicity or antimicrobial (multi)resistance, like ISs or integrons.

Round 2

Reviewer 3 Report

In the opinion of this reviewer, no matter how many loci you compare, I think that some of the isolates analyzed in this study might just be duplicates, and that should be tested. Of course, most of the conclusions of the manuscript still could be valid if the authors answer the issues I raised, but they would be much more solid in turn.
